# The Medicinal Natural Products of *Cannabis sativa* Linn.: A Review

**DOI:** 10.3390/molecules27051689

**Published:** 2022-03-04

**Authors:** Anwuli Endurance Odieka, Gloria Ukalina Obuzor, Opeoluwa Oyehan Oyedeji, Mavuto Gondwe, Yiseyon Sunday Hosu, Adebola Omowunmi Oyedeji

**Affiliations:** 1Department of Chemical and Physical Sciences, Walter Sisulu University, Mthatha 5099, South Africa; endy_odieka@yahoo.com; 2Department of Pure and Industrial Chemistry, University of Port Harcourt, Port Harcourt 500004, Rivers State, Nigeria; gloria.obuzor@uniport.edu.ng; 3Department of Chemistry, Fort Hare University, Alice 5700, South Africa; ooyedeji@ufh.ac.za; 4Department of Human Biology, Walter Sisulu University, Mthatha 5099, South Africa; mgondwe@wsu.ac.za; 5Department of Economics and Business Sciences, Walter Sisulu University, Mthatha 5099, South Africa; yhosu@wsu.ac.za

**Keywords:** medicinal plant, *Cannabis sativa*, phytochemicals, bioactivity, extraction methods, characterization

## Abstract

*Cannabis sativa* is known among many cultures for its medicinal potential. Its complexity contributes to the historical application of various parts of the plant in ethno-medicines and pharmacotherapy. *C. sativa* has been used for the treatment of rheumatism, epilepsy, asthma, skin burns, pain, the management of sexually transmitted diseases, difficulties during child labor, postpartum hemorrhage, and gastrointestinal activity. However, the use of *C. sativa* is still limited, and it is illegal in most countries. Thus, this review aims to highlight the biological potential of the plant parts, as well as the techniques for the extraction, isolation, and characterization of *C. sativa* compounds. The plant produces a unique class of terpenophenolic compounds, called cannabinoids, as well as non-cannabinoid compounds. The exhaustive profiling of bioactive compounds and the chemical characterization and analysis of *C. sativa* compounds, which modern research has not yet fully achieved, is needed for the consistency, standardization, and the justified application of *Cannabis sativa* products for therapeutic purposes. Studies on the clinical relevance and applications of cannabinoids and non-cannabinoid phenols in the prevention and treatment of life-threatening diseases is indeed significant. Furthermore, psychoactive cannabinoids, when chemically standardized and administered under medical supervision, can be the legal answer to the use of *C. sativa.*

## 1. Introduction

The applications of plants as medicines predates human history. A medicinal plant refers to any plant which contains substances of therapeutic potential in one or more of its parts for the synthesis of plant-based drugs [1]. Active medicinal plant ingredients are referred to as bioactive phytochemicals. [2]. These bioactive compounds are believed to increase the ability of plants to survive or adapt to their surroundings [3] and are used as medicines, flavorings, and recreational drugs in humans. One notable medicinal plant that has continued to garner attention over the years, and in recent times, is *Cannabis sativa.*

*Cannabis sativa* L. is known for its medicinal uses since ancient times, because of its rich supply of phytochemicals [4], hence the quest for harnessing its pharmacological potential by scientists. The term “Cannabis” is used to define the products (drugs and essential oils) that are prepared or obtained from the annual herb *C. sativa* and its variants, which are of the family Cannabaceae [5]. The utilization of this multipurpose plant has been restrained for a long time because of the psychoactive effects of a specific cannabinoid (Δ^9^-tetrahydrocannabinol; C_12_H_30_O_2_) [6]. It was strongly prohibited in the twentieth century, and was removed from the British pharmacopeia. The plant was demonized due to its high abuse liability and supposedly insufficient health benefits [7]. Furthermore, due to the inability to prepare standardized preparations, and the diffusion of the recreational use of cannabis below therapeutic concentrations from the end of the 19th to the first half of the 20th century, the medical use of cannabis began to decline [7]. In 1937, the “Marihuana Tax Act”, a federal legislation in the United States, functionally ended all medical uses of cannabis and was removed from the “National Formulary and Pharmacopoeia” in 1941 [7]. In 1961, cannabis resin, extracts, and tinctures were listed in the Schedule I of the single Convention on Narcotic Drugs, which prohibits the use, possession, production, manufacture, export, import, and trade of cannabis, except for medical and scientific purposes [7].

However, in multiple countries today, its cultivation and usage are regulated by laws [6,8]. Recent decriminalization policies and new scientific evidence have increased the interest in the medicinal potential of cannabis and have paved the way for the release of marketing authorizations for cannabis-based products [7]. In 1985, the United States Food and Drugs Administration (US FDA) reconsidered the medical use of cannabinoids, and approved Marinol (dronabinol) and Cesamet (nabilone), two synthetic analogues of tetrahydrocannabinol (THC), for the management of nausea and vomiting associated with cancer chemotherapy [7]. The Office of Medical Cannabis Research (OMC), a Dutch government agency in Europe, became the first organization to obtain the exclusive right to supply medical cannabis to research institutes and pharmacies, and, under the Single Convention on Narcotic Drugs of 1961, to import and export cannabis extracts and resin for medical purposes. Several medical cannabis products, all of which are dried female flowering tops, except Bediol (which is ground into small pieces for its easy manipulation by patients with spasticity), are exported by the OMC, with the proper licenses, to other member states of the European Union [7]. In Italy, the Military Pharmaceutical Chemical Works of Florence became the official national settlement for cultivating and manufacturing medical cannabis with a standard cannabinoid content [7,9]. The Italian Ministry of Health in November 2015, in a Ministerial Degree, authorized the indoor cultivation of cannabis flowering tops at a fixed temperature and at fixed light-dark cycles, leading to a standardized composition of different cannabinoids [9]. Two Italian products (FM1 and FM2) are currently available for consumers, and their use was approved for the treatment of chronic pain, neurological disorders, and other diseases resistant to standard therapies [7,9,10]. Non-psychoactive compounds found in *C. sativa* are associated with fewer side effects and can be used for several industrial applications [6]. The hemp stem supplies both cellulosic and woody fibers. The woody fibers are used for animal beddings, while the cellulosic fibers (bast fibers) are used as a substitute for fiberglass, and to produce bioplastics [4]. Its use as an anti-bacterial finishing agent and in functionalized textiles have also been reported [4]. The inflorescence was used, traditionally, for acute pain, insomnia, coughing, and wounds. The leaves were used for malaria, panting, roundworm, scorpion stings, hair loss, and the greying of hair. The stem bark was used for physical injury and strangury. Vaginal discharge, difficult births, strangury, the retention of the placenta, and physical injuries were treated using the roots [11]. In addition, *Cannabis sativa* contains essential oils of a high value, which can also improve the effectiveness of cannabinoids in pharmaceutical formulations [6].

Despite the influx of chemical-based medicines for treatments, the relevance of medicinal plants in drug development cannot be overemphasized. In recent years, commercial medicinal cannabis products with several variations in the phytocannabinoid content have been licensed and produced in Canada [7,8] and in several other countries. Several synthetic and standardized products are currently available on the market; however, patients’ preferences lean towards herbal preparations, because they are easy to handle and self-administer [7]. Thus, this review intends to highlight the phytochemicals present in the different plant parts, which potentiates their pharmacological activities, as well as the techniques for the extraction, isolation, and characterization of *C. sativa* compounds.

## 2. Methods

Literature on the published works of *Cannabis sativa* was obtained using electronic search engines, such as Google Scholar, the WSU online database (PubChem), and Science Direct. The keywords included, namely, *Cannabis sativa*, medicinal plants, Cannabis phytochemicals, ethnopharmacology bioactivity, and medicinal potentials, were used to source for data. An extensive review of the literature from 2011 through to 2021 (the last ten years) on *Cannabis sativa* L. was used to summarize its medicinal potential. Conversely, an emphasis will be placed on the isolation and characterization techniques from 1970 to 2021 to have a broadened view of the advancements in analytical techniques over the past years. Overall, twenty-nine (29) papers relating to the areas of our focus were chosen and were reviewed by all authors. The results from the search were carefully sorted, based on a general understanding, the review questions, and the related objectives.

## 3. Origin and Botanical Description of *C. sativa*

The genus name *Cannabis* means “cane-like” while *sativa* means “sown”, which signifies that the plant is propagated from the seed and not from the roots [12]. It is believed to have originated in Asia and occurs widely in Africa [12,13]. Central and south-east Asia are the potential natural origins for the domestication of the Cannabis genus [14] and it is known by different common names in different languages (hemp, marihuana, kannabis sativa, ganja, bhang, and al-bhango) [15]. In South Africa, it is colloquially known, in Afrikaans, as “dagga”; in IsiXhoxa as “umfincafincane”; and in Isizulu as “umunyane” [16,17]. Taxonomically, Carl Linnaeus, a Swedish botanist, was the first to coin the name *Cannabis sativa* [18]. Other botanists stated that different types of Cannabis existed based on their size, shape, and resin content (breeding and selection). This review discusses, in particular, *C. sativa*.

The Cannabis phenotype (its observable traits or characteristics, such as its leaf shape and flower color) is based on two main factors: its genetic code (genotype) and the external environmental factors [19].

The roots are branched and are about 30–60 cm deep (Farag and Kayser, 2017) [12]. Cannabis inflorescence is made up of several flower heads found on long leafy stems from each leaf axil. A single brownish fruit, about 2-5mm long, is produced per flower, and it contains a single seed tightly covered with a hard shell [12]. The fruit is propagated by bird and the seed germinates after 8–12 days [18]. The leaves, bracts, and stems of the plant are rich in trichrome, which are a diverse set of structures containing the secondary metabolites (phytocannabinoids and terpenoids) responsible for the defense, plant interactions, and typical smell [18]. Figure 1, below, shows the plant parts of *C. sativa.*

## 4. Phytochemistry of *C. sativa*

### 4.1. Chemical Profile of C. sativa

Cannabis, as a herbal medicine, is a complex mixture of compounds, including cannabinoid phenols, non-cannabinoid phenols (stilbenoids, lignans, spiro-indans, and dihydrophenanthrenes), flavonoids, terpenoids, alcohols, aldehydes, n-alkanes, wax esters, steroids, and alkaloids [6,8,11]. Over 500 chemical compounds have been isolated from the cannabis plant and have been reported [13]. The several classes of secondary metabolites are present in different parts of the plant with a wide range of applications (nutraceuticals, cosmetics, aromatherapy, and pharmacotherapy) that are beneficial for humans. However, previous studies have focused mainly on the cannabinoids, Δ^9^-tetrahydrocannabinol (Δ^9^-THC) and cannabidiol (CBD) in particular; hence, the female flower top is only harvested, while other parts of the plant are discarded [11].

Cannabinoids are a class of terpenophenolic compounds obtained by the alkylation of an alkyl-resorcinol with a monoterpene unit [20,21]. They feature alkyl resorcinol and monoterpene moieties in their molecules [20,22]. This specific chemical class in Cannabis is present in the glandular trichomes, which are abundant in the female flower as phytocannabinoid acids, and in the vegetable matrix as neutral phytocannabinoids [6,13]. They are biosynthesized by the alkylation of olivetolic acid with geranyl-pyrophosphate by a prenyltransferase to produce cannabigerolic acid (CBGA). Decarboxylation, a chemical reaction, converts the acidic forms (Δ^9^ THCA, CBDA, CBCA, and CBGA) into their neutral forms, which are more active and efficient in terms of pharmacological activity [8,23]. To date, 125 cannabinoids have been identified and reported, in addition to five new cannabinoids reported in the past two years, 42 non-cannabinoid phenolics, 34 flavonoids, 120 terpenoids, 3 sterols, and 2 alkaloids [8,11,13]. Terpenoids are the second largest class of cannabis compounds and are responsible for their characteristic aroma [13]. Table 1 below summarizes the classes of compounds isolated from *Cannabis sativa* and the different plant parts in which they are present.

Figure 2, Figure 3 and Figure 4 below show the structures of the different classes of bioactive compounds isolated from *Cannabis sativa* [13,29,41].

### 4.2. Extraction, Isolation, and Chemical Characterization of C. sativa

Many methods have been reported for the extraction of Cannabis in the literature. These include direct maceration (DM), soxhlex extraction, ultrasound-assisted extraction (UAE), supercritical fluid extraction, and microwave-assisted extraction (MAE) [41]. However, two methods of extracting *Cannabis* are differentiated in the literature [41]. The first is the maceration of the plant material in an organic solvent (direct maceration) and the subsequent removal of the solvent by the concentration of the extract under reduced pressure [41]. The second is the innovative supercritical fluid extraction (SFE) method, which involves the use of pressurized solvents [41]. It is necessary for cannabinoid compounds to be extracted with organic solvents instead of water, because the active compounds are less soluble in polar solvents [41]. The most commonly used solvents are ethanol, ether, chloroform, and methanol [42]. When used for extraction, various compounds, including some undesired substances, dissolve together with the cannabinoids [42]. The high solvent power of ethanol for cannabinoid compounds is the reason why it is frequently used in home-made extracts of Cannabis [41]. However, non-desired compounds (chlorophyll, lipids, and waxy materials) are also extracted which, therefore, requires further steps to remove the co-extracted impurities for a high-purity medicinal product to be obtained [41]. A patent on the method for the isolation of herbal and cannabinoid medicinal extracts stated that the solubility of non-therapeutic substances (chlorophyll and waxy materials) is reduced when the solvent is selected from a group that includes acetonitrile, benzene, dichloromethane, diethyl ether, acetone, butanol, ethanol, chloroform, ethyl acetate, hexane, pentane, propanol, tetrahydrofuran, toluene, xylene, and various combinations of these solvents [41]. The International Conference on Harmonization (ICH) recommends the use of less toxic solvents in the manufacture of drug substances and dosage forms, and sets pharmaceutical limits for residual solvents in drug products [43]. Residual solvents pose risks to human health and are classified into three classes. Class 1 solvents (including carbon tetrachloride, benzene, and methyl chloroform) are regarded as human carcinogens and are environmentally hazardous [41]. Class 2 solvents include methanol and hexane, which are generally said to be limited, and they are possible causative agents of irreversible toxicity, such as neurotoxicity or teratogenicity [41]. Class 3 solvents (ethanol and ethyl acetate) are generally regarded as having a low toxic potential to humans [41]). Above all, ethanol is generally recognized as a safe (GRAS) solvent [41]. In a study by Brighenti et al., they compared the following four extraction techniques to obtain a high yield of medicinal cannabinoids: ultrasound-assisted extraction (UAE), microwave-assisted extraction (MAE), supercritical fluid extraction (SFE), and direct maceration (DM). They concluded that DM, with ethanol as the extraction solvent at room temperature for an overall time of 45 min, is the best extraction technique (in terms of a high yield) for non-psychoactive cannabinoids from hemp [44].

Over the last decade, compounds in *Cannabis* have been identified, isolated, and determined by various chromatographic techniques with different spectroscopic detection methods. *C. sativa* samples are analyzed for both legal and medicinal purposes [41]. Nevertheless, the knowledge of their exact composition remains very significant. In 2009, recommended methods for the identification and analysis of cannabis products were released by the United Nations Office on Drugs and Crime [45]. One notable technique that has been employed in identifying the diverse composition of the compounds found is high-performance liquid chromatography (HPLC) [41]. Spectroscopic approaches or methods are based on the variable absorbance or redirection of electromagnetic (EM) radiation by chemical bonds, resulting in the radiation or transition of the sample’s atoms to a higher energy state [46]. Some advantages are attributed to these spectroscopic methods, such as permitting spatial measurements of metabolites and offering a global metabolic fingerprint of a sample with rapid spectral acquisition [46]. Some of these approaches/methods include Fourier transform infrared spectroscopy (FTIR), nuclear magnetic resonance (NMR) spectroscopy, mass spectrometry (MS), HPLC, gas chromatography–mass spectrometry (GC–MS), and liquid chromatography–mass spectrometry (LS–MS) [41,46]. Taking into account the recommended methods and the mandatory requirement of the Ministerial decree to use only chromatographic techniques coupled with mass spectrometric detection, cannabinoid concentrations and its stability in cannabis tea and cannabis oil, prepared from standardized flowering tops obtained from the Military Pharmaceutical Chemical Works of Florence, were studied by Pacifici et al. using easy and fast ultra-high performance liquid chromatography–tandem mass spectrometry (UHPLC–MS/MS) [9,10]. Table 2 is a summary of the reported extraction solvents, as well as the identification, isolation, and characterization methods of *Cannabis.*

### 4.3. Biological Evaluation/Potentials of C. sativa

From the biological point of view, the psychoactive cannabinoids reported include Δ^9^ THC, cannabinol (CBN), and cannabinodiol (CBND), while cannabidiol (CBD) and other cannabinoids are non-psychoactive [8,11]. THC is the major psychoactive component and the toxicity of this metabolite of *Cannabis* is the most studied [11,28]. Its psychoactive component decreases in the order of inflorescence (the flower), leaves, stem, roots, and seeds, respectively [8]. The interest in the potential medical use of cannabis and cannabinoids rose significantly in the 1990s, following the discovery of the endocannabinoid (eCB) system in mammals [7]. The physiological effects of cannabinoids are exerted through various receptors, such as the cannabinoid receptors (CB1 and CB2), adrenergic receptors, and the recently discovered GPCRs (GPR_55_, GPR_3_ and GPR_5_) [8]. Historically, each part of the *Cannabis* plant is indicated mostly for pain killing, inflammation, and for mental illnesses. For example, the *Cannabis* root has been recommended for treating fever, inflammation, gout, arthritis, and joint pain, as well as skin burns, hard tumors, postpartum hemorrhage, difficult child labor, sexually transmitted diseases, gastrointestinal activity, and infections [40]. *Cannabis* has also been used to treat asthma, epilepsy, fatigue, glaucoma, insomnia, nausea, pain, and rheumatism, as well as being used as appetite stimulant and a digestive aid [7,11,13]. Since concentrations above 0.05% are pharmacologically interesting, *Cannabis* inflorescence and leaf material may contain sufficient cannabinoids, mono- and sesquiterpenoids, and flavonoids for therapeutic applications [11]. Cannabis terpenoids and flavonoids, mainly myrcene, limonene, pinene, β-caryophyllene, and cannflavin A, act in synergy with cannabinoids to induce pharmacological effects [7]. It was proven that these compounds, which are synthetized in the aerial parts of the plant, enhance CBD’s anti-inflammatory effects and antagonize THC dysphoric action [96]. Cannabidiol (CBD) and Cannabidavarin (CBDV) (neutral cannabinoids) have been reported to have the therapeutic potential for the treatment of epilepsy (focal seizures), as well as treating nausea and vomiting [97,98]. Conversely, THC and CBN have been found to be active in lowering intraocular pressure, and can be applied in all cases of glaucoma that are resistant to other therapies [9]. Cannflavin A and B are also notable flavonoids (prenylflavonoids) with medicinal potentials, such as their anti-inflammatory, anti-neoplastic, antioxidant, neuro-protective, anti-parasitic, and anti-viral effects [99]. Table 3, below, shows a summary of the reported bioactivities (biological potentials) of the bioactive compounds present in *Cannabis sativa*.

Cannabis female flowering tops can be simply administered through commercially available vaporizers (e.g., Micro Vape, G Pen Herbal Vaporizer, and Volcano), buccal sprays (e.g., Sativex), oral capsules (e.g., Cannador), decoctions, or oils [7]. Only cannabis use through oral or inhalatory administration is allowed. Smoking reduces the bioavailability of cannabis ingredients by 40%, and its complete combustion can cause lung diseases and airway obstructions [7]. Homemade decoctions and pharmacy oils are currently the most widespread cannabis formulations in Europe, making the standardization of preparation difficult [7]. Cannabis pharmacological action is dose-dependent and can induce many adverse effects (AEs), principally related to THC, due to unintentional overdosing [7]. The typical symptoms of cannabis acute intoxication that have been reported are dizziness, confusion, tachycardia, postural hypotension, dysphoria, panic depression, hallucinations, allergic reactions, vomiting, and diarrhea [7,137,138]. Furthermore, withdrawal symptoms, such as irritability, aggression, anxiety, insomnia, decreased appetite, tremors, sweating, and headaches may appear after the abrupt cessation of the long-term administration of high doses of cannabis [7]. According to the ICH efficacy and safety guidelines, it is recommended to start with low doses and increase quantities after a satisfactory period of clinic evaluation, depending on the pharmacological effects and the possible adverse effects [139].

In the current COVID-19 pandemic, scientists are repurposing medicines (identifying new therapeutic use(s) of existing drugs) known for their biological potential (anti-viral or anti-inflammatory properties) to tackle the global issue and similar future viruses [140]. They have hypothesized that CBD could be used as an anti-viral agent [141] or anti-inflammatory [142,143] tool, or to inhibit pulmonary fibrosis in COVID-19 patients [144]. In addition, the known growing evidence of the anxiolytic effects of CBD have also been hypothesized to be used as a therapeutic option to treat long-lasting COVID-19-related anxiety and PTSD [145], which is likely to be a significant issue of the pandemic.

## 5. Conclusions 

With the recent evaluation, acceptance, and legalization of Cannabis products for therapeutic purposes, researchers, particularly in the field of natural products, are challenged to improve and standardize the extraction and characterization of the bioactive compounds from *Cannabis sativa.* Despite various reports of its economic and therapeutic values, it is legal in a handful of jurisdictions (Uruguay, Canada, some US states, and parts of Africa). Presently, Cannabis remains illegal in several countries. This review summarized the biological potential and the techniques for the extraction, isolation, and characterization of *Cannabis sativa* compounds, and it describes the effectiveness of the various parts of the herb in pharmacotherapy. The usage of *C. sativa* roots and stem barks in present-day medical research, and the development of new Cannabis-based medicines or products, in contrast to the flowering part only, is highly recommended because they can be exploited for medicine and other uses. In addition, Cannabis-based pharmaceutical products must undergo long purification processes to eliminate unwanted components such as chlorophyll and residual organic solvents. The use of standardized reagents is also very crucial in the analytical studies of *C. sativa.* Furthermore, future research should seek to clarify the factors responsible for the complexity of *C. sativa* extracts in terms of their chemical compositions, the physical properties of their active ingredients, and their liability to photochemical oxidation.

## Figures and Tables

**Figure 1 molecules-27-01689-f001:**
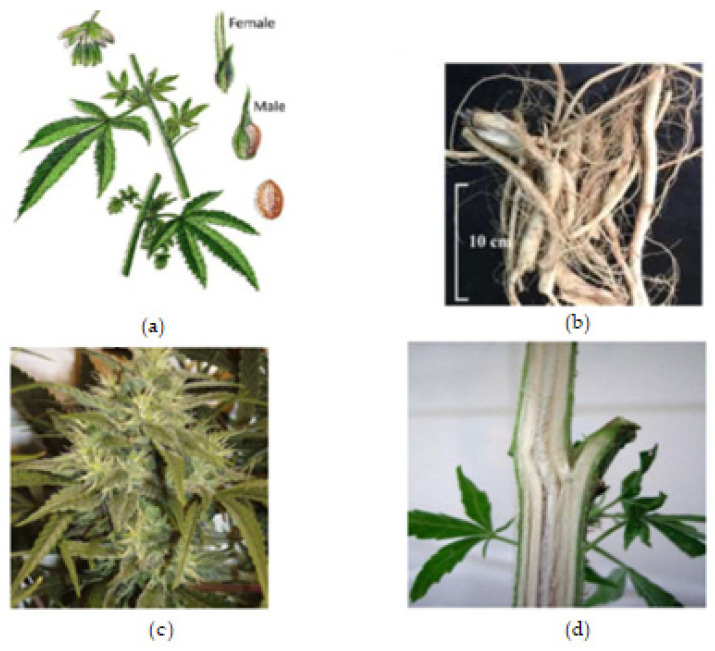
Cannabis plant parts. (**a**) Male and female *Cannabis* flowering parts with fresh leaf and seed. (**b**) Fresh *Cannabis* root. (**c**) Fresh Cannabis inflorescence (flower). (**d**) Fresh *Cannabis* stem bark.

**Figure 2 molecules-27-01689-f002:**
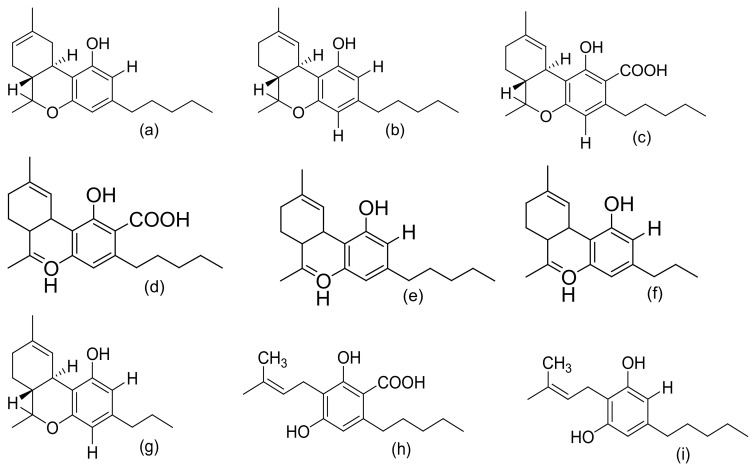
Chemical structures of major cannabinoids; Δ^8^ THC, tetrahydrocannabinol (**a**); Δ^9^-THC, tetrahydrocannabinol (**b**); THCA, tetrahydrocannabinolic acid (**c**); CBDA, cannabidiolic acid (**d**); CBD, cannabidiol (**e**); CBDV, cannabidivarin (**f**); THCV, tetrahydrocannabivarin (**g**); CBGA, cannabigerolic acid (**h**); CBG, cannabigerol (**i**); CBN, cannabinol (**j**); CBNA, cannabinolic acid (**k**); CBC, cannabichromene (**l**); CBCA, cannabichromenic acid (**m**); CBL, cannabicyclol (**n**); CBLA, cannabicyclolic acid (**o**). All structures drawn by Odieka, using ChemDraw Ultra 8.0.

**Figure 3 molecules-27-01689-f003:**
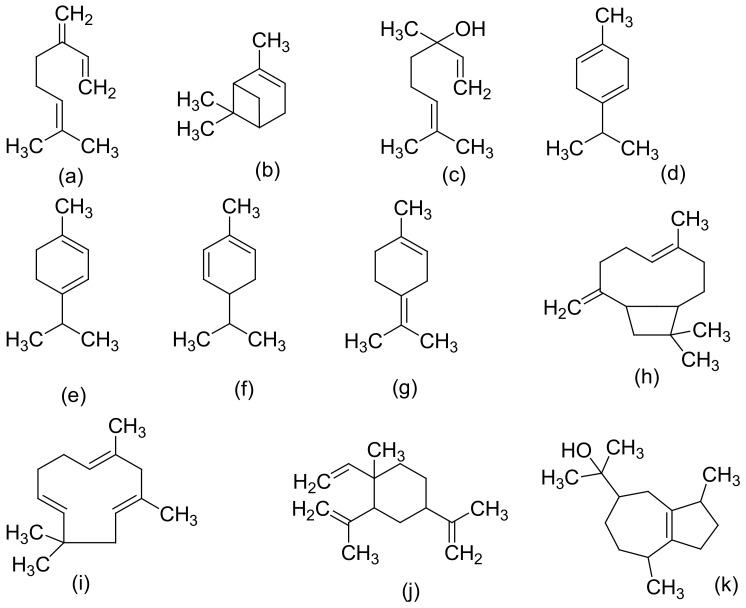
Chemical structures of some *Cannabis sativa* terpenes (Monoterpenes, Sesquiterpenes, and Triterpenoids); Mycene (**a**), α-Pinene (**b**), D-Linalool (**c**), Limonene (**d**), α-Terpinene (**e**), α-Phellandrene (**f**), α-Terpinolene (**g**), β-Caryophyllene (**h**), α-Caryophyllene (**i**), β-Elemene (**j**), Guaiol (**k**), Friedelin (**l**), and Epifriedelanol (**m**). All structures drawn by Odieka, using ChemDraw Ultra 8.0.

**Figure 4 molecules-27-01689-f004:**
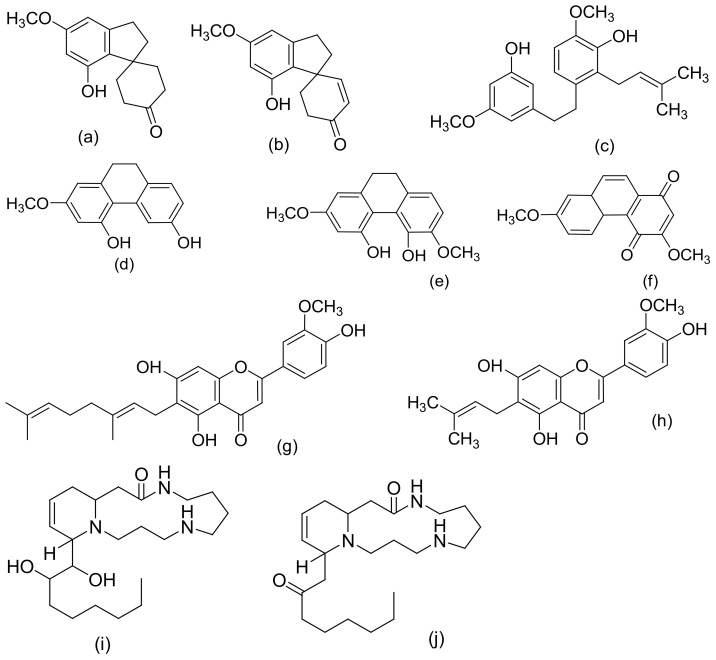
Chemical structures of some non-cannabinoid phenols (Spirans, phenanthrenes, flavonoids, alkaloids); Cannabispiran (**a**), Cannabispirone (**b**), Canniprene (**c**), Cannithrene I (**d**), Cannithrene II (**e**), Debinobin (**f**), Canniflavin A (**g**), Canniflavin B (**h**), Cannabisativine (**i**), and Anhydrocannabisativine (**j**). All structures drawn by Odieka, using ChemDraw Ultra 8.0.

**Table 1 molecules-27-01689-t001:** Chemical compounds Isolated from *Cannabis sativa*.

S/N *	Class of Compounds	Plant Part(s)	Isolated Compounds	References
1	Cannabinoids: -Cannabinoid acids -Neutral cannabinoids -Cannabinoid derivatives -Cannabinoid acid esters	Leaves, flowers, resin, stembarks, and roots	Δ^9^-tetrahydrocannabivarin, α/β-fenchyl Δ^9^-tetrahydrocannabinolate, Δ^9^-tetrahydrocannabinol, α-terpenyl (−)-Δ^9^-*trans*-tetrahydrocannabinolate, γ-eudesmyl (−)Δ^9^-*trans*-tetrahydrocannabinolate, 8α-hydroxy-(−)-Δ^9^-*trans*-tetrahydrocannabinol, 8*b*-hydroxy-(−)- Δ^9^-*trans*-tetrahydro cannabinol, 8-oxo-(−)-Δ^9^-*trans-*tetrahydrocannabinol, Cannabisol, (−)-Δ^9^-*trans*-tetrahydrocannabiphorol, (−)-Δ^9^-*trans*-tetrahydrocannabihexol, (−)-Δ^8^-*trans*-tetrahydrocannabinol, Δ^8^- *trans*-tetrahydrocannabinolic acid, Cannabigerol, 6,7-trans/cis-epoxycannabigerolic acid, Sesquicannabigerol, Cannabigerolic acid, Cannnabigerovarin, Cannabidiol, C_4_-Cannabidiol, Cannabidivarin, C_4_-tetrahydrocannabinol, Cannabichromene, Cannabichromevarin, Cannabichromanone, -D_4_-acetoxycannabichromene, Cannabicitran, Cannabiripsol, Cannabicoumaronone, Cannabifuran, Cannabielsoin, Cannabielsoic acid, Cannabicyclol, Cannabinodiol, bornyl/epi-bornyl-Δ^9^-tetrahydrocannabinlate Cannabinol, Cannabitriol, Cannabimovone, and Cannabioxepane	[8,11,13,24,25,26,27,28,29,30,31]
	**Non-Cannabinoid Constituents**			
2	Non-cannabinoid phenol: -Stilbenoids -Spiro-indans -Phenanthrenes -Lignans, lignanamides, and phenolic amides	Leaves, flowers, stem, hemp pectin, resin, fruit, seed, and root	Dihydrostilbenes, Dihydrophenathrenes, Cannabistilbene, Canniprene, Cannithrene, Denbinobin, Phloroglucinol β-*D*-glucoside, Cannabispiran, Cannabispirone, Cannabispirenone, Cannabispirol, Cannabispirketal, *a*-cannabispiranol-4′-*O*-β-glucopyranose, prenylspirodienone, 4,5-dihydroxy-2,3,6-trimethoxy-9,10-dihydrophenanthrene, Cannabisin A–O 4,7-dimethoxy-1,2,5-trihydroxyphenathrene, 5-methyl-4-pentyl-2,6,2-trihydroxybiphenyl, 5-methyl-4-pentylbiphenyl-2,2,6-triol, *N-trans*-coumaroyltyramine, *N-trans*-feruloyltyramine, *Ntrans*-caffeoyltyramine, 3,3′-demethylheliotropamide, and Grossamide.	[8,13,28,32,33,34]
3	Terpenoids (Terpenes): -Monoterpenes -Sesquiterpenes -Diterpenes -Triterpene,	Essential oils of fresh and dried leaves, flowers, stembarks, and roots	α-pinene, β-pinene, linalool, linalool oxide, myrcene, limonene, camphene, α-terpinene, γ-terpinene, α-terpinolene, α-terpineol, terpinene-4-ol, sabinene, sabinene hydrate, *cis*-sabinene hydrate, α-phellandrene, β-phellandrene, 2-methyl-2-heptene-6-one, borneol, piperitenone, geraniol, carvacrol, carvone, *cis*-carveol, citronellol, bornyl acetate, ipsdienol, germacrene-B, clovandiol, α-bisabolol, β-eudesmol, γ-eudesmol, α-caryophyllene, β-caryophyllene oxide, α-Humulene, Phytol, neophytadiene, friedelin (friedelan-3-one), epifriedelanol, β-amyrin, Vomifoliol, dihydrovomifoliol, and dihydroactinidiolide β-ionone	[8,11,13,28,29,35,36]
4	Flavonoids: -Methylated -Glycosylated (C or O glycosides) -Prenylated -Geranylated	Leaves, flowers, seed, and fruit	Orientin, Vitexin, Isovitexin, Apigenin, Luteolin, Kaempferol, Quercetin, Cytisoside, Cytisoside glucoside, Canniflavone (Cannflavin), Naringenin, and Naringin	[8,11,13,28,29,35,37,38,39]
5	Sterols	Stembarks, roots, and leaves	Campsterol, Stigmasterol, and β- Sitosterol	[11,40]
7	Alkaloids	Roots, leaves, stembark	Cannabisativine and Anhydrocannabisativine	[13,40]
8	Fatty acids: -Saturated and unsaturated fatty acids and their esters	Seeds	Roughanic acid, Stearidonic acid, α-linolenic acid, and oxylipins.	[8,28]
9	Hydrocarbons (*n*-alkane)	-	Δ^9^-Tetrahydrocannabiorcolic acid	[8]

* S/N = Serial Number.

**Table 2 molecules-27-01689-t002:** Reported methods of identification, isolation, and characterization of *C. sativa*.

Extraction Solvent(s)	Matrix and Species	Identification, Isolation, and Purification Methods	Elucidation/Analytical Techniques	Analytes	References
Phenolic Cannabinoids					
Methanol/chloroform mixture.	*C. sativa* inflorescence	Qualitative, quantitative, and comparative derivatization study of cannabinoids	Fast GC–MS	CBDA, CBGA, CBG, CBD, THC, Δ^8^-THC, CBC, THCA, THC	[47]
Supercritical fluid extraction	Plant biomass and medicinal *Cannabis* resin	Quantitative and qualitative analysis of cannabinoids	UHPLC–DAD and statistical analysis	CBDA, THCA, CBD, CBN, CBC, THC	[48]
Methanol/chloroform solvent mixture	*Cannabis* flower samples	Qualitative and quantitative measurement of cannabinoids	HPLC–DAD	Δ^9^-THC, CBD, CBDA, THCA, CBN, CBG, CBGA, Δ^8^-THC	[49]
Ethanol/ethanolic extracts	(i) Lebanese *C. sativa* (ii) *Cannabis* (iii) Decarbo-xylated hemp leaves	(i) Purified by counter-current distribution and silica gel chromatography (ii) Florisil and silica gel column chromatography	GCMS, IR, and ^1^H NMR comparison with an authentic sample	Cannabielsoin, (+)-*trans*-CBT and (−)-*trans* -CBT-OEt-C_5_(Cannibitriol), Cannabicitran, and Monomethylether of CBD	[13,50,51,52]
Sequential extraction (hexanes, CH_2_Cl_2_, EtOAc, EtOH, EtOH/H_2_O, and H_2_O)	(i) Bud and leaves of high-potency variety of *C. sativa*	(i) Silica gel VLC, C_18_-solid phase extraction (SPE), and HPLC (ii) VLC chromatography of hexane extract, TLC, flash silica gel, Sephadex LH-20 chromatography, and semipreparative reversed-phase (RP) and chiral HPLC	HRESIMS, 1D and 2D NMR, GC–MS	*epi*-bornyl Δ^9^-tetrahydrocannabinolate, α-terpenyl Δ^9^-tetrahydrocannabinolate, 4-terpenyl Δ^9^-tetrahydro-cannabinolate, α-cadinyl Δ^9^-tetrahydrocannabinolate, γ-eudesmyl Δ^9^-tetrahydro-cannabinolate, γ-eudesmyl cannabigerolate, 4-terpenyl cannabinolate, bornyl Δ^9^-tetrahydrocannabinolate, α-fenchyl Δ^9^-tetrahydro-cannabinolate, α-cadinyl cannabigerolate, Δ^9^-tetrahydro-cannabinol (Δ^9^-THC), Δ^9^-tetrahydrocannabinolic acid A (Δ^9^-THCA), Cannabinolic acid A (CBNA), and Cannabigerolic acid (CBGA), (±)-6,7-*trans*-epoxycannabigerolic acid, (+)-6,7-*cis*-epoxycannabigerolic acid, (±)-6,7-*cis*-epoxycannabigerol, 5′-Methoxy-cannabigerolic acid, 5′-methyl-4-pentylbiphenyl-2, 2′, 6-triol,7-methoxy-cannabispirone, (±)-6,7- *trans*-epoxycannabigerol, 8α-hydroxy-(−)-Δ^9^-*trans*-tetrahydrocannabinol, 8*b*-hydroxy-(−)- Δ^9^-*trans*-tetrahydro cannabinol, 8-oxo-(−)-Δ^9^-*trans-*tetrahydrocannabinol, 10α-hydroxy-Δ^8^-tetra-hydrocannabinol, 10β-hydroxy- Δ^8^-tetra-hydrocannabinol, 10a-α-hydroxy-10-oxo-Δ^8^-tetrahydrocannabinol, (±)-4-acetoxycannabichromene, (±)-3″-hydroxy-Δ(4″, 5″)-cannabichromene,(−)-7-hydroxycannabichromane,(−)-7 *R*-cannabicoumarononic acid A, 5-acetyl-4-hydroxycannabigerol, 4-acetoxy-2-geranyl-5-hydroxy-3-*n*-pentylphenol, 8-hydroxycannabinol, 8-hydroxycannabinolic acid A, and 2-geranyl-5-hydroxy-3-*n*-pentyl-1, 4-benzoquinone, (±)-4-acetoxycannabichromene, (±)-3”-hydroxy- Δ^4^”-cannabichromene, (–)-7-hydroxycannabichromane, 8-hydroxycannabinol, 8-hydroxy cannabinolic acid, 10*S*-hydroxy-cannabinol, 9*b*,10*b*-epoxyhexahydrocannabinol, 9*a*-hydroxyhexahydrocannabinol, 10*a*-hydroxyhexahydrocannabinol, and 10a*R*hydroxyhexahydrocannabinol	[31,53,54,55]
Hexane extract	(i) *C. sativa* (air-dried and powdered buds) (ii) high-potency variety of *C. sativa* (iii) Illicit Cannabis samples (iv) *C. sativa* inflorescence (strain CINRO) (v) Lebanese *C. sativa* (vi) Hemp (vii) Nepalese and Brazilian C. Sativa.	(i) Column chromatography using silica or alumina, TLC, then fractional distillation and preparative C_18_ HPLC (ii) VLC (vacuum liquid chromatography) silica gel column chromatography, C_18_ HPLC and chiral HPLC (iii) Flash silica gel chromatography (iv) Florisil column chromatography	^1^H NMR, ^13^C NMR (2D NMR) HRESIMS, circular dichroism (CD), UV, LC-HRMS, MS/MS, GC–MS, and confirmation by phytochemical transformations.	Δ^9^-THC, Δ^9^-THC aldehyde, Cannabinoid esters, Cannabisol, Δ^9^-*trans*-tetrahydrocannabiphorol, Δ^9^-*trans*-tetrahydrocannabihexol, Cannabidiorcol, Cannabidihexol (CBDH) and Cannabidiphorol (CBDP), Cannabitwinol, Cannabinodivirin and cannabinodiol (CBND), Cannabichromene (CBC), 9a-hydroxyhexahydrocannabinol, 7-oxo-9a-hydroxyhexa-hydrocannabinol, 10a-hydroxyhexahydrocannabinol, and 10a-*R*-hydroxyhexahydrocannabinol	[53,56,57,58,59,60]
Ethyl acetate extracts	*Cannabis* resins, tinctures, and leaves	-	GC–MS and GC–FID analysis	Δ^9^-THC and Δ^9^-THCA	[61]
Petroleum ether	(i) Cannabis tincture of Pakistani origin (ii) Brazilian *C. sativa* (iii) Cannabis leaves and flowers (Maryland and Czechoslovakian origin) (iv) Congo *C. sativa* (v) Hashish and Cannabis sativa	(i) Silicic acid column chromatography (ii) Silica gel and Florisil chromatography, preparative TLC	IR, NMR, MS, GC–MS confirmed by synthesis	Δ^9^-THCV, Δ^9^-THCO or Δ^9^-THC, Δ^8^-THC (Δ^8^-THCA), Cannabielsoin acid A (CBEAA,), Cannabielsoin acid B, and Cannabicyclovarin (CBLV)	[61,62,63,64,65,66,67]
Benzene	(i) Fresh *C. sativa* leaves from Thailand (ii) Fresh tops and leaves of *C. sativa* (iii) Hemp	Polyamide and silica gel column chromatography	IR, UV, NMR, and comparing UV spectrum with that of derivatives	Δ^9^-THCVA, CBDV, THCV, CBCV, Cannabigerovarin CBGV, cannabigerovarinic acid (CBGVA), CBDA, cannabidivarinic acid (CBDVA), Cannabicyclolic acid (CBLA), and Cannabichromenic acid (CBCA)	[49,68,69]
Acetone extract	(i) Leaves of *C. sativa* (Mexican strain) (ii) Wax of decarboxylated aerial parts of *C. sativa* (Carma strain) (iii) Cannabis variety (carmagnola)	(i) Silica gel column chromatography (ii) Silica and alumina column chromatography, followed by normal phase (NP)-HPLC. (iii). Flash chromatography, over reverse-phased C_18_ silica gel followed by normal-phase HPLC	FAB–MS, ^1^H-NMR, ^13^C-NMR), and ESI–MS semisyn-thesis.	Cannabigerolic acid (CBGA), dihydroxycannabigerol derivative (camagerol), Sesquicannabi-gerol, Cannabimovone, and Cannabioxepane	[70,71,72,73]
Essential/volatile oils					
Methanol dilutions	*Cannabis sativa* oil samples	Separation/quantitation of cannabinoids	Fast-GC–FID	CBD, CBN, and THC	[74]
Essential oil	Fresh *C. sativa* L. from India	Fractional distillation and chromatography over alumina.	GC–MS and physico–chemical analyses	α-terpinene, β-phellandrene, γ-terpinene, α-terpinolene, α-pinene, β-pinene, camphene, linalool, α-terpineol, terpinene-4-ol, linalool oxide, and sabinene hydrate	[13]
Volatile/essential oils	(i) *Cannabis* (Dutch and Turkish) (ii) Fresh leaves of *Cannabis sativa* and *Cannabis indica*	(i) Hydrodistillation or through nitrogen extraction (ii) Hydrodistillation, steam distillation, and supercritical fluid extraction	Capillary gas chromato-graphy, GC–MS analysis	cis-β-ocimene, trans-β-ocimene, α-phellandrene, D_3_-carene, Δ^4^-carene, sabinene and α-thujene, caryophyllene, humulene, *trans*-β-bergamotene, *cis*-β–farnesene, δ–limonene, carophyllene oxide, linalool, *trans*-α- bergamotene, *cis*- β -farnesene, menthol, eucalyptol, and Carvone.	[13,75]
Essential oil	Cannabis (marijuana fresh and dried buds)	Steamdistillation	GC–MS and GC–FID	Ipsdienol, *cis*-carveol, and *cis*-sabinene hydrate	[76]
Essential oil	*C. sativa* resin	Minor terpenic component analysis	GC–MS and GC retention time	α-gurjunene, α-bisabolol, α-cedrene, α-cubebene, δ-cadinene, epi-β-santalene, farnesol, γ-cadinene, γ-elemene, γ-eudesmol, guaiol, (E,E)-α-farnesene, (*Z*)-β-farnesene, and farnesyl acetone	[77]
Essential oil	*Cannabis*	Steam distillation and silica gel chromatography	GC, GC–MS	eugenol, methyleugenol, iso-eugenol, trans-anethol, and *cis*-anethol (simple phenols)	[78]
Essential oil	*C. sativa*	Column chromatography of the essential oil	GC and GC–MS analyses	Iso-caryophyllene, β-selinene, selina-3,7(11)-diene, and selina-4(14),7(11)-diene	[13,79]
Non-cannabinoid phenols					
Ethanol/ethanolic extract	(i) South African *Cannabis* variant (ii) Saudi Arabia hashish (iii) Leaves of *C. sativa* (iv) High-potency *Cannabis* variety grown in Mississippi (v) Hemp pectin (vi). *Cannabis* roots (vii) Roots, stem, and leaves of a Mexican variant of *Cannabis* sativa	(i) Partitioning and chromatography on silica and polyamide columns (ii) Normal and reversed phase chromatographic techniques (iii) Purification by macro reticular resin, silica gel column chromatography, and Sephadex-LH-20 (iv) TLC, chromatography over alumina, and recrystallization (v) Partitioning and TLC eluted with chloroform:acetone:ammonia (1:1:1) (vi) Series of acid–base extractions and silica-gel chromatography followed by crystallization of the alkaloid from acetone	IR, GCMS, UV, 1D NMR (1H NMR, 13C NMR) and 2DNMR (COSY, HSQC, HMBC, and ROESY), ESI–MS, comparison with authentic samples, X-ray crystallography, and semi-synthesis	β-cannabispiranol, *b*-cannabispirol, 5-hydroxy-7-methoxyindan-1-spiro-cyclohexane, 7-hydroxy-5-methoxyindan-1-spiro-cyclohexane, and 5,7-dihydroxyindan-1-spiro-cyclohexane, Cannabispirketal and the glycoside, *a*-cannabispiranol-4′-*O*-β-glucopyranose, 3,4′,5-trihydroxy-dihydrostilbene, 4,5-dihydroxy-2,3,7-trimethoxy-9,10-dihydrophenanthrene, 4-hydroxy-2,3,6,7-tetramethoxy-9,10-dihydrophenanthrene and 4,7-dimethoxy-1,2,5-trihydroxyphenanthrene, Rutin, friedelin (friedelan-3-one) and epifriedelanol, Anhydrocannabisativine and cannabisativine, α,α′-dihydro-3′,4,5′-trihydroxy-4′-methoxy-3-isopentenylstilbene, α,α′-dihydro-3,4′,5-trihydroxy-4-methoxy-2,6-diisopentenylstilbene, α,α′-dihydro-3′,4,5′-trihydroxy-4′-methoxy-2′,3-diisopentenylstilbene, α,α′-dihydro-3,4′,5-trihydroxy-4,5′-diisopentenylstilbene, and combretastatin B-2	[32,33,55,80,81,82,83,84]
Benzene	Dried leaves of Japanese cannabis	Chromatographed on a polyamide column followed by silica gel chromatography	IR, ^1^H NMR, MS, UV.	Cannabispirol and acetyl Cannabispirol	[85]
Acetone	*C. sativa* (CARMA)	Gravity column chromatography on silica gel and purified by crystallization from ether and HPLC	Identified according to its physical and spectroscopic properties and synthesis	Debinoben (1,4-phenanthrenequinone)	[86]
Sequential extraction (Hexane, EtOAc, CH_2_Cl_2_, EtOH, EtOH/H_2_O, and H_2_O	High-potency variety of *C. sativa* (Mississippi)	VLC, silica gel column chromatography, and RP–HPLC	1D and 2D NMR, IR analysis	acetoxy-6-geranyl-3-*n*-pentyl-1,4-benzoquinone, 4,5-dihydroxy-2,3,6-trimethoxy-9,10-dihydrophenanthrene, 4-hydroxy-2,3,6,7-tetramethoxy-9,10-dihydrophenanthrene, 4,7-dimethoxy-1,2,5-trihydroxyphenanthrene, Cannflavin C and β-sitosteryl-3-*O-*β-d-glucopyranoside-2′-*O*-palmitate, α-cannabispiranol, Chrysoeriol, 6-prenylapigenin, and Cannflavin A and β-acetyl cannabispiranol	[55]
Hexane extract	Leaves of *Cannabis sativa*	Isolation by normal-phase chromatography followed by C_18_-HPLC	NMR and ESI–MS analysis	Prenylspirodinone and 7-*O*-methyl-cannabispirone	[55,87]
Dichloro-methane extract	(i) Decarboxy-lated *C. sativa* hemp (ii). Thai *Cannabis sativa* leaves (iii) Panamanian variety of cannabis	C_18_ flash chromatography, followed by silica gel gravity column chromatography and HPLC	HR–ESIMS and NMR (^1^H, ^13^C, HSQC, and HMBC) data, X-ray crystallography, and confirmation by hydrogenation	Isocannabispiradienone and *a*-Cannabispiranol, Cannabispira-dienone, and Cannabidihydro-phenanthrene (Cannithrene1 and Cannithrene2)	[88,89]
Methanol/methanolic extract	(i) Branches and leaves of hemp (ii) Pollen grains of Mexican variety of *C. sativa* (iii) Dried leaves of South African and Indian cannabis sativa (iv) Panamanian variant of *C. sativa* (v) Leaves and branches of *C. sativa*	(i) TLC, silica gel column chromatography, normal-phase preparative HPLC, and Sephadex LH-20 column chromatography (ii) Partitioning, silica gel chromatography, Sephadex LH-20 chromatography, semi preparative LC	MS, 1D and 2D NMR, UV experiments, IR, X-ray crystallography and confirmation by total synthesis	Rutin, Quercetin-3-*O*-α-*L*-rhamnoside, kaempferol-3-*O*-α-L-rhamnoside, apigenin-7-*O*-α-L-rhamnoside, apigenin-7-*O*-β-*D*-glucopyranoside, luteolin-7-*O*-β-*D*-glucopyranoside, 1,3,6,7-tetrahydroxyl-2-C-β-*D*-glucopyranosyl xanthone, vitexin, orientin, apigenin-6,8-di-C-β-*D*-glucopyranoside, vitexin-2″-*O*-α-*L*-rhamnoside, orientin-2″-*O*-β-*D*-glucopyranoside, quebrachitol, inositol and uracil, kaempferol-3-*O*-sophoroside (196) and quercetin-3-*O*-sophoroside, cannabispirone; cannabispirenone, Cannabispiran, Isocannabispiran Canniprene, Cannabistilbene I and Cannabistibene II, and 2,3,5,6-tetramethoxy 9,10-dihydrophenanthrenedione	[90,91,92,93,94]
Mixture of hydro-alcoholic and organic solvents	*C. sativa* inflorescence (Ferimon, Uso-31, Felina 32, and Fedora 17 cultivars)	Metabolic and chemical profiling to identify and quantify compounds of different classes	NMR, GC–MS, UHPLC, and HPLC–PDA	Sugars, organic acids, amino acids, cannabinoids, terpenoids, phenols, tannins, flavonoids (Quercetin, Naringenin, and Naringin) and biogenic amines	[35,37]
Diethyl ether	Stem exudate (greenhouse-grown *C. sativa)*	TLC and acid hydrolysis of the exudate	^1^H NMR and GC–MS	Phloroglucinol β-*D*-glucoside	[95]

**Table 3 molecules-27-01689-t003:** Summary of reported bioactivities associated with isolated compounds and essential oils from *Cannabis sativa*.

Isolated Bioactive Compound	Bioactivity/Uses	References
Tetrahydrocannabinol THC	Antioxidant, anti-pruritic, and anti-inflammatory effects	[29,100]
Cannabidiol CBD	Anti-convulsive, anti-inflammatory, immunosuppressive properties, antioxidant, and anti-psychotic effects	[101,102,103]
Cannabigerol CBG	Anti-fungal effects, anti-cancer, anti-depressant, mild anti-hypertensive agent, analgesic, and anti-erythemic effects	[29,104]
Cannabichromene CBC	Anti-inflammatory and analgesic	[29]
Cannabinol CBN	Sedative, anti-convulsant, anti-inflammatory, antibiotic, and anti-MRSA activity	[29]
Tetrahydrocannabivarin THCV	Anti-convulsant	[29]
Tetrahydrocannabinolic acid THCA	Immunomodulatory, anti-inflammatory, neuroprotective, anti-neoplastic activity, and antiemetic effects	[29,105]
Cannabidavarin CBDV	Anti-convulsant (anti-epileptic) properties and anti-emetic properties	[106,107]
Cannabidiolic acid CBDA	Anti-emetic effects	[104,108,109,110]
β-Myrcene	Anti-inflammatory and analgesic sedative agent	[7,29,111]
*D*-Limonene	Strongly anxiolytic, anti-depressant, antibiotic, and anti-cancer agent	[7,29]
β-Ocimene	Anti-convulsant activity, anti-fungal activity, anti-tumor activity, and pest resistance	[112,113]
γ-Terpinene	Anti-inflammatory activity, antioxidant, and anti-proliferative activity	[114,115]
α-Terpinene	Antioxidant	[29]
α-Pinene	Anti-inflammatory, bronchodilator, anti-microbial, and anxiolytic effects	[7,11,29,116]
Linalool	Analgesic and anticonvulsant, anxiolytic, anti-depressant, anti-glutamatergic, anti-leishmanial activity, anticancer agent, anti-nociceptive, and anti-depressant effects	[111,117,118,119]
α-Phellandrene	Anti-nociceptive, anti-depressant, anti-arthritic and allergic, and anti-hyperalgesic effects	[120,121,122]
Terpinolene	Anti-fungal and larvicidal, anti-nociceptive, anti-inflammatory antioxidant, and anti-cancer effects	[123,124]
β-Caryophyllene	Cardio-protective, hepato-protective, gastro-protective, neuro-protective, nephro-protective, antioxidant, anti-inflammatory, anti-microbial, anti-pruritic, and immunomodulatory activities	[7,125,126,127]
Caryophyllene Oxide	Anti-fungal, insecticidal/anti-feedant, and anti-platelet effects	[29]
β-Elemene	Anti-cancer and anti-tumor	[128]
Guaiol	Anti-inflammatory, antioxidant, anti-cancer anti-rheumatic, antiseptic, diaphoretic, diuretic, and laxative effects	[29,129]
Friedelin	Anti-inflammatory, anti-pyretic, and anti-tuberculosis agent	[130,131]
Epifriedelanol	Antioxidant	[132]
Cannflavin A and B	Anti-inflammatory, neoplastic, antioxidant, neuroprotective, anti-parasitic, and anti-viral agent	[4,7,29,99]
Apigenin	Anxiolytic and estrogenic properties, anti-tumor, antioxidant, anti-inflammatory, anti-osteoporosis, and immune regulation effects	[4,133]
Vitexin and Isovitexin	Antioxidant, anti-cancer, anti-inflammatory, anti-diabetic, anti-microbial, anti-viral, anti-hyperalgesic, and neuroprotective effects	[134]
Quercetin	Anti-cancer/anti-proliferator, antioxidative/anti-aging, anti-viral, anti-inflammatory, cardio-protective, skin-protective, anti-coagulant, and anti-platelet effects	[135]
Luteolin	Neuroprotective effects, anti-inflammatory, and antioxidant effects	[136]
Lignans	Antioxidant, anti-viral, anti-diabetic, anti-tumorigenic, and anti-obesity activities	[4]

## Data Availability

Not applicable.

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
