# Peer review of "The Medicinal Natural Products of *Cannabis sativa* Linn.: A Review"

_molecules, 2022, doi:10.3390/molecules27051689_

Round 1
Reviewer 1 Report
This review aims to highlight the biological and rural livelihood potentials of the plant parts, the techniques for extraction, isolation and characterization of C. sativa compounds including a unique class of terpenophenolic compounds called cannabinoids as well as non-cannabinoid compounds. The use of C. sativa is still limited and illegal in most countries and regions. Thus, this review will attract interests to many pharmaceutical and medical experts.
It is an interesting and acceptable study, but requires some minor modifications to make it suitable for publication.
- Page 7-9, Figure 2.1- 2.3: the structure of compounds from Cannabis sativa should be redrew in consistent style like ACS 1996.
- Page 11, table 2: the table 2 was hard to read, and need be revised and re-designed in a suitable style for readers to read concisely.
- The English level of this manuscript should be proved by professional language editor.
Reviewer 2 Report
General comments
The authors of this review paper aimed to conduct a review on Cannabis sativa L. with the stated objective to "highlight the phytochemicals present in the different plant parts which potentiate pharmacological activity and rural livelihood as well as the techniques for extraction, isolation and characterization of C. sativa compounds." This objective itself is disjointed given the very different subject matter of phytochemistry/analysis and rural economics. While an important matter, the inclusion of rural livelihood into this review seemed out of place. Further, this topic was allotted a short subsection in this review article. The importance of cannabis cultivation in the economic livelihood of smallholding farms and the implication of burdensome regulation and licensing fees probably warrants an entirely separate review. It is recommended that the authors find a way to integrate this subject into the current manuscript or remove it entirely. If the authors choose the latter, they should expand this section and revise the introduction to incorporate a rationale for including this topic in their review. Additionally, several extraneous statements are made throughout this paper that make the narrative seem too broad and unfocused.
Specific comments
1. INTRO
Line 35-37: This statement needs to be revised or removed. Clinical trials routinely provide conflicting evidence regarding the effectiveness of medicinal plants. The word "verified" is too strong of a word. The article referenced by the authors (Sofowora et al., 2013) does not provide a source for the statement that "over 90% of traditional medicine recipes/remedies contain medicinal plants" and is therefore not reliable as a reference for this.
Line 38-40: This statement is extraneous to the stated focus of this review paper. The term "phytochemical" is well-established in the literature making it unnecessary to include a formal definition here.
Line 40: Typo. "These bioactive compounds in plant are" ... is this meant to be plural (i.e. "plants") or stated as "in planta"?
Line 41: "...overcome challenges..." be more specific or remove from the statement.
Line 43: Whole plants, seeds, etc. are used to season foods. I'm unfamiliar with plant bioactives that are extracted and then used as a flavour additive.
Line 48: It is not necessary to write out the full scientific name each time it is mentioned; use C. sativa following the first mention.
Line 49-51: Remove "Although"; also, change "have been restrained" to "has been restrained"; describe how long cannabis usage has been considered illicit by authorities; include "in multiple countries today" after the word "However".
Line 52-53: "The other compounds contained in the plant are very useful with less side effects and hence used for several industrial applications [6]." Revise to "Non-psychoactive compounds found in C. sativa are associated with fewer side effects and can be used for several industrial applications."
Line 64: Relevance to what?
Line 67: Please revise "some" to "several".
Line 69: It is unclear how this subject ties into the other subjects of this review (pharmacological activity, extraction techniques), and a rationale for focusing on this is not provided in the introduction.
2. METHODS
Line 72: Remove "s" from "Literatures" (from line 76 as well).
Line 73: Please explicitly state all databases from which information was sourced, electronic or otherwise.
Line 76: Please use the full scientific version of the plant name at first mention (on line 45) and then abbreviate to C. sativa L. thereafter. Why did the authors choose to review papers onwards from 2011?
Line 77-78: Change to past-tense (i.e. "will be" to "was") since this data collection for this review has been completed.
Line 79-80: "...as well as the shift in framework needed to derive beneficiation for all the stakeholders." The meaning of this statement is unclear. The introduction should be developed more so that the connection to the authors' stated focus on rural livelihood becomes clear.
Line 80-81: Please expand on methods … how many researchers were involved? Was any literature excluded and if so, why? How was the literature "carefully sorted"?
3. ORIGIN AND BOTANICAL DESCRIPTION OF CANNABIS SATIVA
Line 82-113: Section 3 should be reduced in length substantially or removed completely. In-depth historical information seems outside the scope of your thesis, as defined in the introduction. This makes your narrative seem unfocused. The authors should remove factoids that are extraneous to your central objective and zero in on what is relevant to phytochemicals, extraction techniques, and rural livelihood. Consider what part of this is relevant to your thesis statement and what will be helpful for your audience. For example, the parts of the plant shown in Figure 1 is useful since you will be making reference to the various plant parts throughout this paper. Information on the number of leaflets and sepals in Cannabis leaves and flowers is distracting. Also, for consistency, the plant name should be shortened to C. sativa in the section title.
4. PHYTOCHEMISTRY OF C. SATIVA
Line 136: What is meant by "a wide range of applications"? Please clarify whether the authors are referring to the role secondary metabolites play in planta or the applications they may have for human use.
Line 137: The word "dwelt" is incorrectly used here.
Line 143: Change "are" to "is.
Line 147: The abbreviations mentioned here need to be defined first.
Line 148: Efficient in what sense?
Table 1: It is redundant to describe cannabinoids as "phenolic" as non-phenolic cannabinoids do not exist. Please revise. Section 4 (Flavonoids) are a type of non-cannabinoid phenolic and thus should be reclassified in your table to fall under section 2. Please define in the table footnote what "S/N" means.
Figure 2.1: This figure needs to be modified as some elements are sized incorrectly. For example, (j) and (k) appear stretched, while (l)-(o) are much larger than (a)-(i). Similar attention to detail should be paid to Figures 2.2 and 2.3 so that sub-elements are uniform.
Line 196-197: Why focus on these two methods of extraction? The authors should briefly mention what the other methods are and why they are inferior. The word "reported" is used twice in this sentence.
Line 197-199: Please provide a name for this extraction technique.
Line 203-204: The authors have included chemical structures of all cannabinoid chemical classes. Why are these solvents used for extraction?
Line 205-206: Please name a couple of these non-desired substances. Are these non-desirables the same as the non-therapeutic substances mentioned in line 209? Please clarify.
Line 216: Insert the word "are" before "regarded".
Line 218: "limited" in what sense?
Linen 221: Remove "to man".
Line 22: Put GRAS acronym before the word "solvents".
Line 224: The abbreviation for supercritical fluid extraction was introduced in line 200.
Line 226: Best in terms of what? Yield? Preservation of therapeutic compounds? Please be specific.
Line 228: For clarity, please insert the word "spectroscopic" between the words "different" and "detection", if that indeed is what the authors are referring to.
Line 231-233: This sentence is long-winded. Also, change "have" to "has" and revise one of the instances of the word "technique".
Line 239: The way this list is written is confusing. In lines 236-238, the authors outline the advantages of spectroscopic approaches for detection. Thin layer and column chromatography are not spectroscopic methods. These technologies are used to separate mixtures into individual compounds and can be coupled to a spectroscopic detector in order to identify these compounds. Please revise to clarify.
Table 2: The second column ("Matrix & Species") should be revised as the sole focus of this manuscript is on a singular species (C. sativa). Perhaps the authors are referring to cultivar? There are several regional references (e.g. "Lebanese hashish", "Nepalese" and "Brazillian" marijuana). This is too vague. The authors should indicate the cultivar name or coding associated with each (this can be found in the methods section of the references they have cited).
Line 253: Change "content" to "component".
Line 254: How can something be the only and the most? Please remove "only". Also, please remove "makes it" as the connection between its psychoactive activity and the number of studies associated with it is not clear.
Line 259: Remove "releasing".
Line 279-281: What is meant by the statement "repurposing medicines"? They should refer to cannabis specifically.
Section 4.4. needs to be revised significantly for grammar.
Line 289: Change "A lot have..." to "A lot has" and provide some references for this statement.
Line 290: It would be helpful if the authors explain why scarcity of research in this area is a problem.
Line 291-292: Unclear what is meant by "world's drug Cannabis". Has research among smallholders been hindered in most or all producing countries? If the answer is "most" then the authors should describe in more detail what the existing research says.
Line 293-298: This statement should be revised to be more clear and specific.
Line 298-303: The authors should describe the research they reference here. Considering this is one of the main topics of the review paper, details on this subject are sparse in this section.
Line 301: The phrases "Still and all" and "who produces what" are informal for an academic paper and should be revised.
Line 326-328: Are these recommendations based on an economic study of cannabis production in rural settings, or are these personal recommendations by the authors? If so, what are they based on? Perhaps the authors can include some references here.
5. CONCLUSIONS AND RECOMMENDATION
Line 341: Please clarify that in some jurisdictions the sale of Cannabis products for therapeutic purposes is legal. It remains illegal in several countries at present.
Line 345: Revise "concludes on the effectiveness" to "describes".
Line 346-349: Why is research on other components of the plant highly recommended by the authors?
Line 352-354: "Furthermore, the complexity of C. sativa extracts in terms of chemical composition, physical properties of their active ingredients, and liability to photochemical oxidation demands further research." ... what do the authors mean by this? This recommendation is vague. What about this should future research seek to clarify specifically?
Line 354-357: "The paper also examined the importance of factoring the sustainable livelihood of smallholder farmers and small cottage/entrepreneurial development in both legal and business frameworks by the removal of high cost of license and over-regulation which may be area specific." ... the authors did not examine the impact of the high cost of licensing and over-regulation. They merely stated it. They need to provide more evidence from the literature.
Line 357-358: These topics were not discussed in the body of this manuscript, making this recommendation seem like it was randomly inserted into the conclusion.
Reviewer 3 Report
The review presented summarized the botanical and phytochemical properties of Cannabis sativa. Medical applications and economic potentials of Cannabis sativa were also discussed briefly. The reviewer finds the manuscript pleasing to read in general. But as a scientific publication, revisions and more thinking are required.
- All the chemical structures need to be redrawn using a scientific software, e.g., ChemDraw or counterparts, for a more scientific and accurate rendering. Just name a few: bonds are not of equal length; stereochemistry is not consistent.
- In terms of medicinal potentials, reviewer believe that authors mean ‘anti-neoplastic’ when mentioned ‘neoplastic’.
- For the conclusion, authors advocate ‘The legal framework should ensure the removal of overburdened cost of license and over-regulation that will drive smallholder farmers and small entrepreneurial Cannabis enterprises away from the sector’. Reviewer is not arguing against the statement. But the conclusion is not convincing without a thorough discussion and solid data, especially the negative physiological and socioeconomic impacts of Cannabis sativa cannot be neglected.
Round 2
Reviewer 2 Report
The authors have edited the body of their manuscript sufficiently; however, I believe the title requires amending as "Livelihood prospects" is no longer relevant to the subject matter in this article.
